# Pharmacophoric-constrained heterogeneous graph transformer model for molecular property prediction

Yinghui Jiang[1,7], Shuting Jin [1,2,3,7], Xurui Jin[1,7], Xianglu Xiao[1], Wenfan Wu[1], Xiangrong Liu[2,3], Qiang Zhang [4], Xiangxiang Zeng[5], Guang Yang [6✉] & Zhangming Niu [1,6✉]

Informative representation of molecules is a crucial prerequisite in AI-driven drug design and discovery. Pharmacophore information including functional groups and chemical reactions can indicate molecular properties, which have not been fully exploited by prior atom-based molecular graph representation. To obtain a more informative representation of molecules for better molecule property prediction, we propose the Pharmacophoric-constrained Heterogeneous Graph Transformer (PharmHGT). We design a pharmacophoric-constrained multi-views molecular representation graph, enabling PharmHGT to extract vital chemical information from functional substructures and chemical reactions. With a carefully designed pharmacophoric-constrained multi-view molecular representation graph, PharmHGT can learn more chemical information from molecular functional substructures and chemical reaction information. Extensive downstream experiments prove that PharmHGT achieves remarkably superior performance over the state-of-the-art models the performance of our model is up to 1.55% in ROC-AUC and 0.272 in RMSE higher than the best baseline model) on molecular properties prediction. The ablation study and case study show that our proposed molecular graph representation method and heterogeneous graph transformer model can better capture the pharmacophoric structure and chemical information features. Further visualization studies also indicated a better representation capacity achieved by our model.

[1] MindRank AI Ltd., 310000 Hangzhou, China. [2] School of Informatics, Xiamen University, 361005 Xiamen, China. [3] National Institute for Data Science in Health and Medicine, Xiamen University, 361005 Xiamen, China. [4] School of Informatics, Zhejiang University, 310013 Hangzhou, China. [5] School of Information Science and Engineering, Hunan University, 410082 Changsha, Hunan, China. [6] National Heart and Lung Institute, Imperial College London, London, UK. [7] These authors contributed equally: Yinghui Jiang, Shuting Jin, Xurui Jin. ✉email: g.yang@imperial.ac.uk; zhangming@mindrank.ai

The goal of drug discovery is to find novel molecules with desired properties, and predicting the properties of molecules accurately has been one of the critical issues. The key step of molecule properties prediction is how to represent the molecules that map the molecular information to a feature vector. Conventional methods learn the representations depending on the expert-crafted physic-chemical descriptors[1], molecular fingerprints[2], or the quantitative structure-activity relationship (QSAR) method[3,4].

In recent decades, deep learning methods have shown strong potential to compete with or even outperform conventional approaches. Graph neural networks (GNNs) have gained increasing more popular due to their capability of modeling graph-structured data. For the association prediction task of biological network data, the heterogeneous graph neural network algorithms[5–7] have achieved remarkable results. Molecules can be naturally expressed as a graph structure, so the GNNs method can effectively capture molecular structure information, including nodes (atoms) and edges (bonds)[8]. Compared with the conventional methods, deep learning methods can use SMILES or molecular graph as input which is more informative and lead to significant improvement in downstream tasks such as molecules properties prediction. The graph-based molecular property prediction models view the molecules as graphs and use graph neural networks (GNN) to learn the representations and try to capture the topological structure information from atoms and bonds. Due to their ability to represent molecules as graphs, they are an important research area for molecular property prediction tasks. The most representative GNNs, including GCN[9–14], GAT[15–17], and MPNN[18–20] etc., have been actively used in the field of molecular graphs-based for molecular properties prediction. However, these models ignore the information of fragments that contain functional groups. Recently, Zhang et al.[17,21] works have begun to focus on molecular fragment information to predict the properties of molecules.

Although incorporating fragment information into graph architectures to benefit some molecular property estimation tasks has attracted research attention in recent years, there still are two issues that impede the usage of GNNs in this field: (1) those models have not provided a global chemical perspective method to better integrate atom and fragment information and both ignore the reaction information between fragments; (2) lacking the generalization ability of the different types and feature dimensions of atoms, fragments, and bonds. To address those two issues, more comprehensive information from different levels needs to be embedded and there is still a demand to develop a heterogeneous GNNs model for molecular property prediction.

In the study, we propose a Pharmacophoric-constrained Heterogeneous Graph Transformer model (PharmHGT) to comprehensively learn different views of heterogeneous molecular graph features and boost the performance of molecule property prediction (the code is available on GitHub: https://github.com/mindrank-ai/PharmHGT). Firstly, we use the reaction information of BRICS to divide the molecule into fragments that contain functional groups and retain the reaction information between these fragments to construct a heterogeneous molecular graph containing two types of nodes and three types of edges (Fig. 1). Then, to comprehensively consider the multi-view and multi-scale graph representations of molecules and the reaction information connecting the fragments, we propose a novel heterogeneous graph transformer model based on message passing. Specifically, we use two variants of transformers to learn the features of edges and nodes in heterogeneous graphs respectively, and aggregate and update these features of edges and nodes through message passing to obtain the representation of heterogeneous molecular graphs. Extensive experiments show that the model has outperformed the advanced baselines on multiple benchmark datasets. Further ablation experiments also showed the effectiveness of learning molecules representation from different perspectives. Our contributions can be summarized as follows:

- We obtain the pharmacophore information from the compound reaction and retain the reaction information between the fragments. On this basis, a heterogeneous molecular graph representation method is constructed.
- We develop a heterogeneous graph transformer framework, which is able to efficiently capture the information of different node types and edge types, including the reaction information between fragments through the fusion of multi-views information of heterogeneous molecular graphs.
- We evaluate PharmHGT on nine public datasets and demonstrate its superiority over state-of-the-art methods.

## Results and discussion
In this section, we present the related work of this field and the proposed PharmHGT model. We also presented the results of PharmHGT for molecular property prediction on ten datasets, these experiments datasets are from Wu et al.[22], including four classification and three regression tasks. More descriptions of the data process can be found in Supplementary Note.

**Related work.** For graph data, if there is only one kind of node and one kind of connection relationship from one node to another, it is called a homogeneous graph, otherwise, it is a heterogeneous graph. Currently, most of molecular graph is based on homogeneous graph and using the heterogeneous graph to learn the representation is still blank. In this section, we review related prior homogeneous graph-based molecular representation methods and heterogeneous graph embedding. We focused on the homogeneous graphs that have some relevance to our model and those models were used for baseline comparison with PharmHGT.

*Fragment-based homogeneous graph-based molecular representation.* It has been demonstrated that many characteristics of molecules are related to molecular substructures which contain functional group information. Zhang et al.[17] obtained two fragments by breaking the acyclic single bonds in a molecule and exploited a fragment-oriented multi-scale graph attention network for molecular property prediction (FraGAT), which first proposed the definition of molecule graph fragments that may contain functional groups. However, FraGAT directly adopted the Attentive FP[15] to get molecular graph embeddings, and the obtained fragments by this method also is rough because there are multiple substructures in one molecular. Zhang et al.[21] proposed the Motif-based Graph Self-supervised Learning (MGSSL) model, which designed a molecule fragmentation method that leverages a retrosynthesis-based algorithm BRICS and additional rules for controlling the size of motif vocabulary and used GNNs to capture the rich structural and semantic information from graph motifs. However, this work still did not consider the reaction information between substructures obtained through BRICS and effectively combine the information of the atom and the substructures. Nevertheless, these works still prove that considering more information from molecular substructures with functional groups can provide a more informative representation that can significantly improve the performance of downstream tasks, but how and which kinds of information to be embedded needs more exploratory work.

*Message passing neural networks.* Gilmer et al.[18] proposed Message Passing Neural Networks (MPNNs), which is the first general framework for supervised learning on graphs, and can

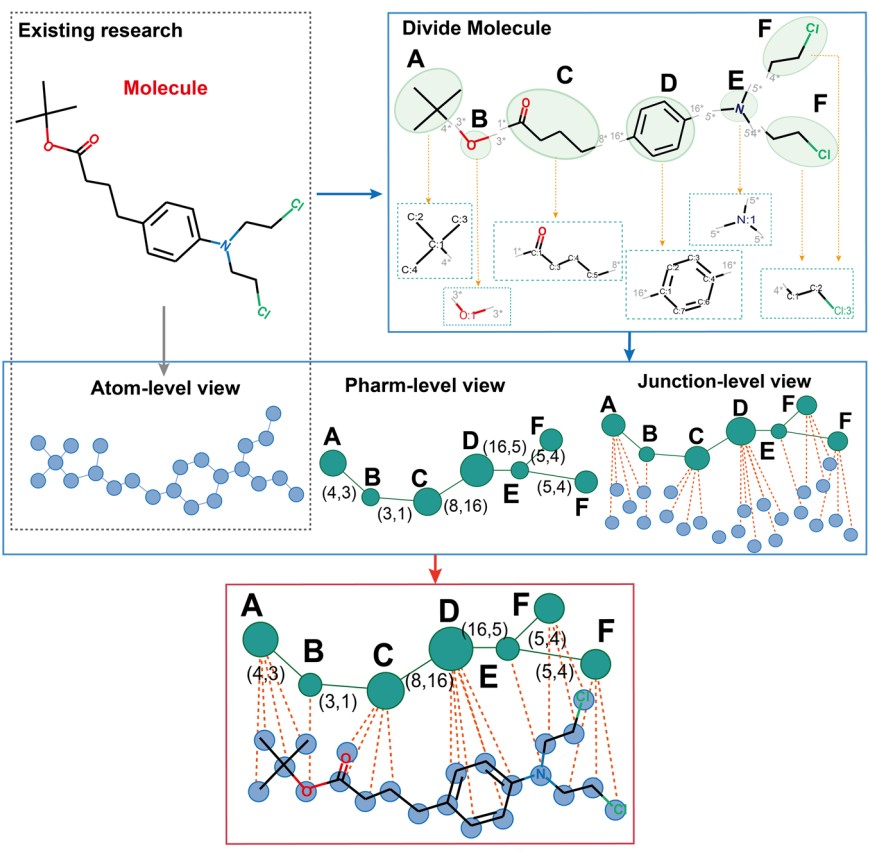

**Fig. 1 An example of the overview of the molecular segmentation process and the construction of the heterogeneous molecular graph.** In the heterogeneous molecular graph at the bottom, the green nodes represent fragments with pharmacophore information, and the blue nodes represent the atoms of the molecule. The green edges are the reaction information between fragments, the red dotted line edges are the related information of the atoms that connect the fragments, and the edges between atoms are bonds.

effectively predict the quantum mechanical properties of small organic molecules. The MPNNs framework is capable of learning the representations from molecular graphs directly. Many researchers made improvements on this basis and proposed many models based on MPNN. Yang et al.[23] introduced a graph convolutional model, called Directed MPNN (D-MPNN), which used messages associated with directed bonds to learn molecular representations. Song et al.[19] proposed a directed graph-based Communicative Message Passing Neural Network (CMPNN) that comprehensively considered the information of atoms and bonds to improve the performance of molecular properties prediction. However, those MPNNs have ignored the chemical reactions information, which is vital for molecular properties from the knowledge of chemistry and pharmacy.

*Transformer architecture.* Researchers proposed the Transformer architecture eschewing recurrence and convolutions entirely and instead based solely on the attention mechanism[24]. Ying et al.[25] have explored several simple coding methods of the graph, mathematically showing that many popular GNN variants are actually just special cases of Graph transformers. In the field of representation learning of molecules, Rong et al.[8] proposed Graph Representation frOm self-superVised mEssage passing tRansformer (GROVER), which can learn the rich structure and semantic information of molecular from a large amount of unlabeled molecular data. Chen et al.[26] proposed the Communicative Message Passing Transformer (CoMPT), which reinforces message interactions between nodes and edges based on the Transformer architecture.

**Heterogeneous graph-based molecular representation.** In the field of recommendation systems, heterogeneous graph models are popular for mining scenarios with nodes and/or edges of various types[27–29]. Heterogeneous graphs are notoriously difficult to mine because of the bewildering combination of heterogeneous contents and structures. The current representation learning for molecules is still at the level of homogeneous graphs, but in addition to the basic atom-based molecular graph representation, some fragment-based and motif-based representation schemes have been proposed to represent a molecule. Obviously, if these representation schemes can be constructed for a comprehensive heterogeneous molecular graph representation, it will be more conducive to capturing the characterization of molecules and potentially improve the performance of downstream tasks. In this paper, we propose a new molecular heterogeneous graph construction method and propose a heterogeneous graph transformers model that can efficiently learn molecular representations.

**Overview of PharmHGT architecture.** The key idea of PharmHGT is additionally capturing the pharmacophoric structure and chemical information feature from the heterogeneous molecular graph. Generally, the heterogeneous graph is associated with the node and edge attributes, while different node and edge types have unequal dimensions of features. As shown in Fig. 2, the proposed PharmHGT consists of three major modules: multi-view molecular graph construction, aggregation of nodes and edges information by heterogeneous graph transformer, and the

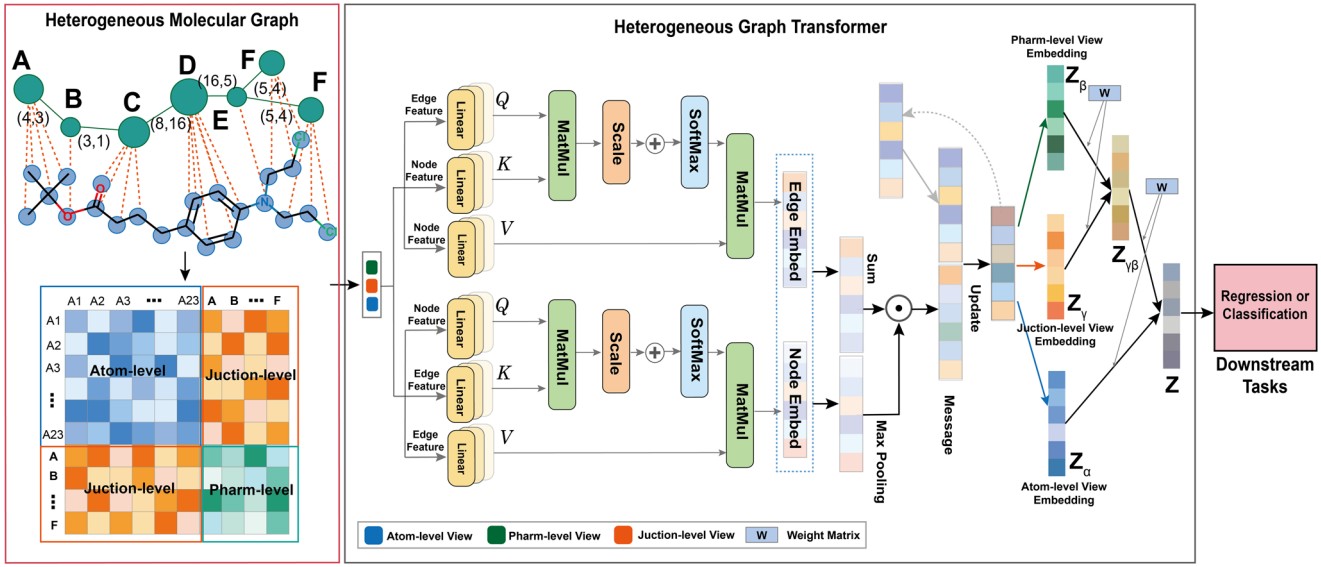

**Fig. 2 Illustration of Pharmacophoric-Constrained Heterogeneous Graph Transformer for molecular property prediction (PharmHGT).** Firstly, the heterogeneous molecular graph is formalized as the feature matrix. Then, the feature matrix of each view will first do message passing independently to obtain the graph feature matrix of each view. Finally, the junction_view feature matrix will first do attention aggregation with the pharm_view feature matrix to obtain the aggregation feature matrix, then that matrix will do attention aggregation with the atom_view feature matrix, and finally, obtain the features of each node, and input those nodes as a sequence into the GRU to get the representation vector of the entire small molecule.

attention mechanism to integrate multi-view molecular graph features for molecular property prediction.

### Experiments

*Datasets.* In order to better compare and prove the effectiveness of PharmHGT, we select nine benchmark molecular datasets for experiments including Blood-brain barrier permeability (BBBP), BACE, ClinTox, Tox21, SIDER, and HIV for classification tasks, and ESOL, Freesolv and Lipophilicity for regression tasks. Below, we include a brief introduction of these datasets.

- *Classification tasks.* The BBBP dataset has 2035 molecules with binary labels of permeability properties, which are often used to predict the ability of molecules to penetrate the blood-brain barrier. The BACE dataset has 1513 molecules, which provides quantitative ($IC_50$) and qualitative (binary label) binding results for a set of inhibitors of human $\beta$-secretase 1 (BACE-1). The ClinTox dataset has 1468 approved drug molecules and a list of molecules that failed due to toxicity during clinical trials. The Tox21 dataset has 7821 molecules for 12 different targets relevant to drug toxicity and was originally used in the Tox21 data challenge. The SIDER dataset has 1379 approved drug molecules and their side-effect, which are divided into 27 system organ classes. The HIV dataset has 41127 molecules and these molecules are tested for their ability to inhibit HIV replication.
- *Regression tasks.* The ESOL dataset records the solubility of 1128 compounds. The FreeSolv includes a total of 642 molecules are selected from the Free Solvation Database. The Lipophilicity dataset provides the experimental result of octanol/water distribution coefficient (logD at pH 7.4) of 4198 compounds.

*Implementation details.* Following the previous works, we illustrate the results of each experiment with 5-fold cross-validation and replicate training five times to increase the credibility of our model. All benchmark datasets have been split as training,

validation, and test sets with a ratio of 0.8/0.1/0.1, while all models were evaluated on random or scaffold-based splits as recommended by[23]. The node and edge features are processed by the open-source package RDKit, and the detail is demonstrated in Supplemental Experimental Procedures (Tables S1–S5).

**Baselines.** In the study, we compare our model with eight baseline methods including 3 types.

- **Fragment-based method**: The AttentiveFP[15] is a graph neural network architecture, which uses a graph attention mechanism to learn from relevant drug discovery datasets. The FraGAT[17] exploited a fragment-oriented multi-scale graph attention network for molecular property prediction; The MGSSL[21] designed Motif-based Graph Self-supervised Learning (MGSSL) by introducing a novel self-supervised motif generation framework for GNNs.
- **MPNN baselines**: The MPNN[18] abstracts the commonalities between several of the most promising existing neural models into a single common framework, and focused on obtaining effective vertices (atoms) embedding by message passing module and message updating module; The DMPNN[23] used messages associated with directed bonds rather than those with vertices; The CMPNN[19] introduced a new message booster module to rich the message generation process.
- **Graph transformer baseline**: The CoMPT[26], with a Transformer architecture, has learned a more attentive molecular representation by reinforcing the message interactions between nodes and edges; The GROVER model[8] standard for Graph Representation frOm self-superVised mEssage passing tRansformer, which can learn rich structural and semantic information of molecules from enormous unlabeled molecular data by carefully designed self-supervised tasks in node-level, edge-level, and graph-level. In addition, the Graphormer model is also based on graph transformer, but Graphormer is a 3D model, which requires the 3D conformation of each small molecules. There are some inconsistencies between our model and

**Table 1 Overall performance comparison to the state-of-the-art methods on molecular property prediction classification tasks.**

**Classification (ROC-AUC%, higher is better↑)**

| Dataset | BBBP | BACE | ClinTox | Tox21 | SIDER | HIV |
|---|---|---|---|---|---|---|
| **Molecules** | **2039** | **1513** | **1478** | **7831** | **1427** | **41127** |
| **Task** | **1** | **1** | **2** | **12** | **27** | **1** |
| **Splitting strategy** | **Scaffold** | **Scaffold** | **Scaffold** | **Scaffold** | **Scaffold** | **Scaffold** |
| AttentiveFP | 90.8 (5.01) | 78.4 (0.02) | 93.3 (2.04) | 80.7 (2.02) | 60.5 (6.01) | 75.7 (1.40) |
| FragGAT | 92.3 (4.04) | 80.1 (0.86) | 93.9 (2.06) | 82.3 (1.62) | 63.3 (3.23) | 76.1 (0.65) |
| MGSSL | 69.7 (0.91) | - | 80.7 (2.12) | 76.5 (0.31) | 61.8 (0.81) | - |
| MPNN | 91.3 (4.14) | 77.9 (1.62) | 87.9 (5.25) | 80.8 (2.39) | 59.5 (3.03) | 74.1 (1.15) |
| DMPNN | 91.9 (3.04) | 80.9 (0.60) | 89.7 (4.01) | 82.6 (2.32) | 63.2 (2.28) | <u>78.6 (1.40)</u> |
| CMPNN | 92.7 (0.23) | 82.1 (0.64) | 90.2 (1.20) | 80.6 (1.57) | 61.6 (0.31) | 77.4 (0.50) |
| CoMPT | 93.8 (2.13) | 81.9 (1.26) | 93.4 (1.85) | 80.9 (1.40) | 63.4 (2.97) | 78.1 (2.60) |
| GROVER$_{base}$ | 93.6 (0.80) | <u>82.6 (0.70)</u> | 92.5 (1.30) | 81.9 (2.00) | 65.6 (0.60) | 62.5 (0.90) |
| GROVER$_{large}$ | <u>94.0 (1.90)</u> | 81.0 (1.40) | <u>94.4 (2.10)</u> | <u>83.1 (2.50)</u> | <u>65.8 (2.30)</u> | 68.2 (1.10) |
| PharmHGT | **95.4 (1.15)** | **86.5 (2.21)** | **94.5 (0.42)** | **83.9 (0.56)** | **66.9 (1.63)** | **80.6 (0.21)** |

The results of baselines are obtained by us using a 5-fold cross-validation with scaffold split and doing experiment on each task for one time. The values in this table are the Mean and standard deviation of ROC-AUC values. The best performance is marked in bold and the second best is underlined to facilitate reading.

Graphormer in terms of target tasks and inputs. For better comparability, we have not added Graphormer into the benchmark, but we give the computational results in the Supplemental Experimental Procedures (Table S7 and Table S8).

**Performance comparison**

*Performance in classification tasks.* Table 1 presents the area under the receiver operating characteristic curve (ROC-AUC) results of eight baseline models on six classification datasets. The Clintox, Tox21, ToxCast, and SIDER are all multi-task learning tasks, including total of 658 classification tasks. Compared with traditional baselines and several GNN-based models, PharmHGT achieved large increases of ROC-AUC in all datasets (we give the prediction ROC curved plots in Fig. S1 and Fig. S2). PharmHGT is designed to be more attentive to pharmacophores, which makes this model more explainable. To note, the PharmHGT outperformed the pre-train methods with less computational cost. We also give computing resources performance comparison to the state-of-the-art methods base on ESOL datasets, see the Table S6.

*Performance in regressions tasks.* Solubility and lipophilicity are basic physical chemistry property, which is vital for explaining how molecules interact with solvents and cell membrane. Table 2 compares PharmHGT results to other state-of-the-art model results. The best-case RMSE of the PharmHGT model on ESOL, FreeSolv and Lipophilicity are $0.680 \pm 0.137$, $1.266 \pm 0.239$, and $0.583 \pm 0.063$ in random split, and $0.839 \pm 0.049$, $1.689 \pm 0.516$ and $0.638 \pm 0.040$ in scaffold split. These results indicate that better representations of molecular graphs containing more information could significantly increase the model performance on downstream tasks.

**Ablation study**. We conducted ablation studies on PharmHGT to explore the effect of atom-level view, pharm-level view, and junction-level view. Under the same experimental setup, we implement seven simplified variants of PharmHGT on the two benchmarks:

- (1) PharmHGT_$\alpha$: by only retaining the atom-level graph.
- (2) PharmHGT_$\beta$: by only retaining the pharm-level graph with reaction information.

- (3) PharmHGT_$\gamma$: by only retaining the junction-level graph.
- (4) PharmHGT_$\beta\alpha$: by aggregating features of the pharm-level graph with reaction information to the atom-level graph.
- (5) PharmHGT_$\gamma\alpha$: by aggregating features of the junction-level graph to the atom-level graph.
- (6) PharmHGT_$\beta\gamma$: by aggregating features of the pharm-level with reaction information to the junction-level graph.
- (7) PharmHGT_$\gamma\alpha\beta$: by aggregating features of the junction-level graph with the atom-level graph, then to the pharm-level graph.

As shown in Fig. 3, the PharmHGT considering the heterogeneous feature information from all views shows the best performance among all architectures. The exclusions of the atom-level, pharm-level, or junction-level view both caused decreases in performances and the PharmHGT_$\beta$ performs the worst when only retaining the pharm-level graph with reaction information. It indicates that lacking information from the atoms can not effectively represent the characteristics of the molecule. When combining two kinds of feature information, PharmHGT_$\gamma\alpha$ aggregates the junction-level graph into an atom-level graph and it has the best performance among the models with one or two views. It proves that integrating the feature information from molecular fragments can improve the prediction performance. The results of PharmHGT demonstrate that further integrating the information from the reaction can obtain the most effective molecular characterization.

**Representation visualization**. To investigate the molecular representations learning ability of PharmHGT, we used t-distributed Stochastic Neighbor Embedding (t-SNE) with default hyper-parameters to visualize molecular representations of the Tox21 dataset in Fig. 4. For this result, we define all molecules with a label of 0 as non-toxic compounds, and any molecule with a label of 1 as a toxic compound, and molecules with similar toxicity tend to have more similar feature spaces. Therefore, we visualize their embeddings by t-SNE and evaluate whether the model can learn effective molecular representations by whether the toxic and non-toxic molecules have a clear boundary. The DMPNN has second performance in Tox21 task

**Table 2 Overall performance comparison to the state-of-the-art methods on molecular property prediction regression tasks.**

Regression (RMSE, lower is better↓)

| Dataset | ESOL | FreeSolv | Lipophilicity | ESOL | FreeSolv | Lipophilicity |
|---|---|---|---|---|---|---|
| Molecules | 1128 | 642 | 4200 | 1128 | 642 | 4200 |
| Tasks | 1 | 1 | 1 | 1 | 1 | 1 |
| Splitting strategy | Random | Random | Random | Scaffold | Scaffold | Scaffold |
| AttentiveFP | 0.853 (0.060) | 2.030 (0.420) | 0.650 (0.030) | 0.877 (0.029) | 2.073 (0.183) | 0.721 (0.001) |
| FragGAT | 0.878 (0.124) | 1.538 (0.640) | 0.645 (0.042) | 0.884 (0.041) | 2.065 (0.201) | 0.750 (0.013) |
| MPNN | 1.167 (0.430) | 2.185 (0.952) | 0.672 (0.051) | 1.541 (0.630) | 2.430 (0.821) | 0.730 (0.063) |
| DMPNN | 0.980 (0.258) | 2.177 (0.914) | 0.653 (0.046) | 1.050 (0.008) | 2.182 (0.183) | 0.683 (0.016) |
| CMPNN | 0.789 (0.112) | 2.007 (0.442) | 0.614 (0.029) | 0.845 (0.039) | 1.833 (0.580) | 0.658 (0.029) |
| CoMPT | 0.774 (0.058) | 1.855 (0.578) | 0.592 (0.048) | 0.915 (0.042) | 1.959 (0.808) | 0.646 (0.028) |
| GROVER$_{base}$ | 0.888 (0.116) | 1.592 (0.072) | 0.660 (0.061) | 1.185 (0.160) | 2.001 (0.081) | 0.817 (0.008) |
| GROVER$_{large}$ | 0.831 (0.120) | 1.544 (0.397) | 0.643 (0.030) | 1.098 (0.178) | 1.987 (0.072) | 0.823 (0.010) |
| PharmHGT | **0.680 (0.137)** | **1.266 (0.239)** | **0.583 (0.026)** | **0.839 (0.049)** | **1.689 (0.516)** | **0.638 (0.040)** |

The results of baselines are obtained by us using a 5-fold cross-validation with scaffold split or Random split and doing experiments on each task for one time. The values in this table are the Mean and standard deviation of RMSE values. The best performance is marked in bold and the second best is underlined to facilitate reading.

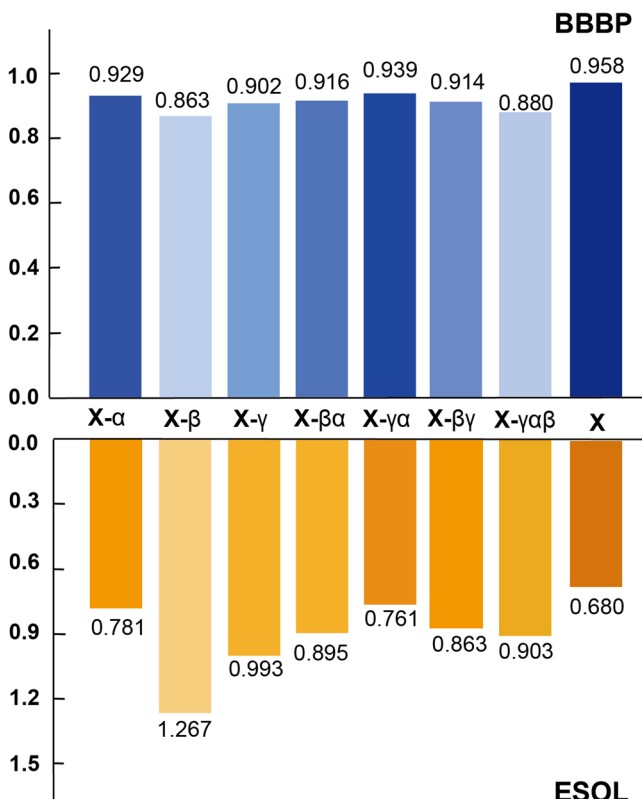

**Fig. 3 Ablation results on BBBP and ESOL datasets.** The "**X**" represent the PharmHGT, the "**X**_" represents different PharmHGT variants of aggregating atom-level, junction-level, and pharm-level features.

and achieves reasonable distinction between toxic and non-toxic molecules (Fig. 4a), however, PharmHGT shows a more visible boundary to classify toxic and non-toxic compounds (Fig. 4c). In addition, the single-view (Fig. 4b) performance is far inferior to the multi-view PharmHGT (Fig. 4c), which also proves the necessity of considering the molecular multi-view information.

**Case study**. Pharmacophore is a molecular framework that defines the necessary components that are responsible for specific properties. Accordingly, identifying and adding the pharmacophore structure information associated with the target property into the model is vital for molecular representation. To illustrate the pharmacophore structure learning ability of PharmHGT, we visualize molecular features on the ClinTox dataset and select six molecules that are toxic in clinical trials and several of them have been applied in the clinical setting as chemotherapeutic drugs. The toxicities of these six molecules are highly correlated with the contained pharmacophore (i.e., some specific sub-structure). Figure 5c shows that our PharmHGT can aggregate molecules with similar toxic pharmacophores together and distinguish them from non-toxic samples; PharmHGT_$\alpha$ cannot well aggregate molecules with similar toxic pharmacophores, and have limited discrimination from negative samples without the pharm-level view (Fig. 5b); The pretraining model Grover, which achieves second performance in ClinTox subtask, can only aggregate only a few molecules with similar toxic pharmacophores (Fig. 5a) and the discrimination for non-toxic samples is far less than PharmHGT. This indicates that the embedded representations learned by PharmHGT can capture functional group structural information more effectively.

## Conclusions

In this paper, we propose PharmHGT, a pharmacophoric-constrained heterogeneous graph transformer model for molecular property prediction. We use the reaction information of BRICS to decompose molecules into several fragments and construct a heterogeneous molecular graph. Furthermore, we develop a heterogeneous graph transformation model to capture global information from multi-views of heterogeneous molecules. Extensive experiments demonstrate that our PharmHGT model achieves state-of-the-art performance on molecular properties prediction. The ablation study and case study also demonstrate the effectiveness of using pharmacophore group information and heterogeneous molecules information of molecules.

## Methods

**Notation and problem definition**. We use the BRICS[30] to decompose molecules into several fragments with pharmacophore, and retain the reaction information between fragments to construct a heterogeneous molecular graph. The heterogeneous molecular graph is denoted $G = \{V, E\}$, the $G$ associated with a node type mapping function $\varphi : V \to \mathcal{O}$ and an edge type mapping function $\psi : E \to \mathcal{P}$, where $\mathcal{O}$ and $\mathcal{P}$ represent the set of all node types and the set of all edge types, respectively. We treat molecular structure as heterogeneous graphs to capture the chemical information from functional substructures and chemical reactions. We propose three views of molecular graph representation schemes, which are the

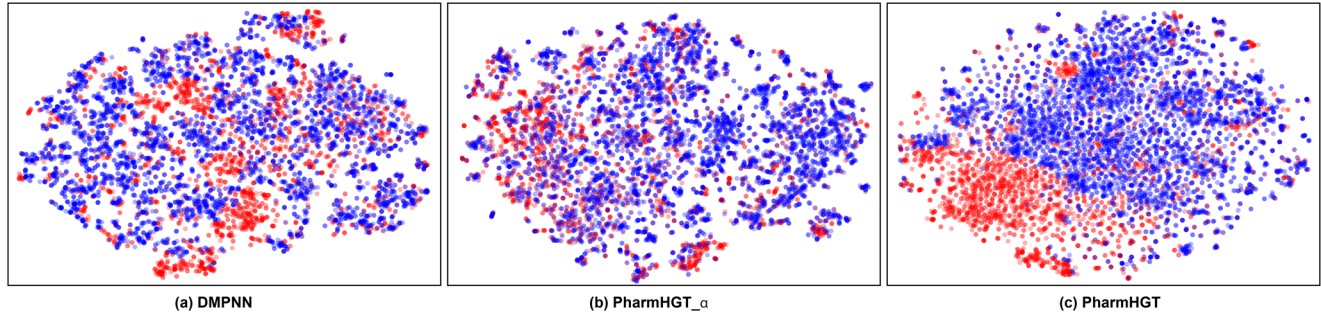

**Fig. 4 Visualization of molecular features.** Visualization of molecular features for Tox21 from **a** DMPNN, **b** PharmHGT_α, and **c** PharmHGT with t-SNE. All molecules with a label of 0 as non-toxic compounds, and any molecule with a label of 1 as a toxic compound, where toxicity compounds are colored in red and the non-toxic ones are in blue.

atom-level view, pharm-level view containing pharmacophore information as well as reaction information, and junction-level view to comprehensively represent a molecule (Fig. 1). The specific definition is as follows:

**Definition 1.** (Atom-level view.) An atom-level view can be denoted as graph $G^\alpha = (V^\alpha, E^\alpha)$, for each atom $v_i^\alpha$ we have $v_i^\alpha \in V^\alpha$ where $1 \le i \le |N^\alpha|$ and $|N^\alpha|$ is the total number of atoms, while for each bond $e_{ij}^\alpha$ we have $e_{ij}^\alpha \in E^\alpha$ where $1 \le i, j \le |N^\alpha|$. For featurization, the $V^\alpha$ is represented as $X_v^\alpha \in \mathbb{R}^{N^\alpha \times D_v^\alpha}$ where $D_v^\alpha$ is the dimensions of atom features, the $E^\alpha$ is represented as $X_e^\alpha \in \mathbb{R}^{M^\alpha \times D_e^\alpha}$ where $|M^\alpha|$ is the total number of directed bonds, $D_e^\alpha$ is the dimensions of bond features.

**Definition 2.** (Pharm-level view.) A pharm-level view can be denoted as graph $G^\beta = (V^\beta, E^\beta)$, for each pharmacophore $v_i^\beta$ we have $v_i^\beta \in V^\beta$ where $1 \le i \le |N^\beta|$ and $|N^\beta|$ is the total number of pharmacophores, while for each BRICS reaction type $e_{ij}^\beta$ we have $e_{ij}^\beta \in E^\beta$ where $1 \le i, j \le |N^\beta|$. For featurization, the $V^\beta$ is represented as $X_v^\beta \in \mathbb{R}^{N^\beta \times D_v^\beta}$ where $D_v^\beta$ is the dimensions of pharmacophore features, the $E^\beta$ is represented as $X_e^\beta \in \mathbb{R}^{M^\beta \times D_e^\beta}$ where $|M^\beta|$ is the total number of BRICS reaction types, $D_e^\beta$ is the dimensions of BRICS reaction type features.

**Definition 3.** (Junction-level view.) A junction-level view can be denoted as graph $G^\gamma = (V^\gamma, E^\gamma)$, for each node $v_i^\gamma$ we have $v_i^\gamma \in V^\gamma$ where $1 \le i \le |N^\gamma|$ and $|N^\gamma|$ is the total number of atoms and pharmacophores, while for each edge $e_{ij}^\gamma$ we have $e_{ij}^\gamma \in E^\gamma$ where $1 \le i, j \le |N^\gamma|$. For featurization, the $V^\gamma$ is represented as $X_v^\gamma \in \mathbb{R}^{N^\gamma \times D_v^\gamma}$ where $D_v^\gamma$ is the dimensions of pharmacophore features, the $E^\gamma$ is represented as $X_e^\gamma \in \mathbb{R}^{M^\gamma \times D_e^\gamma}$ where $|M^\gamma|$ is the total number of atoms and pharmacophores junction relationships, $D_e^\gamma$ is the dimensions of junction relationship information.

An example of the heterogeneous molecular graph and its multi-view is illustrated in Fig. 1, which contains 2 node types and 3 edge types. Given the above definitions, our main task is to learn representations of heterogeneous molecular graphs.

**Overview of PharmHGT.** The key idea of PharmHGT is additionally capturing the pharmacophoric structure and chemical information feature from heterogeneous molecular graphs. Generally, the heterogeneous graph is associated with node and edge attributes, while different node and edge types have unequal dimensions of features. The framework consists of three parts: multi-view molecular graph construction (Fig. 1), aggregation of nodes and edges information by heterogeneous graph transformer, and the attention mechanism to integrate multi-view molecular graph features for molecular property prediction (Fig. 2).

**Obtaining the embedding of nodes and edges.** The inputs of PharmHGT are the feature matrix of node $X_V$ and the feature matrix of edge $X_E$, the features of all nodes can be obtained according to the intensity of the attention between the node and the related edge. The multi-head self-attention mechanism enhances the signal of the node in each view. Specifically, the basic block of PharmHGT is the usual attention module:

$$[Q, K, V] = h(X)[W^Q, W^K, W^V] \tag{1}$$

where $h(X)$ is the hidden features, $W^Q, W^K, W^V$ are the projection matrices. The normal attention module is the dot product self-attention, the $Q, K, V$ is considered in the same semantic vector space, which is not adapted in heterogeneous graph. Therefore, we build a multi-view attention function to get more information from different views, and the function can be formulated as:

$$\text{Attention}(Q, K, V) = \sum_{p \in P} \Omega^p \sigma\left(\frac{Q^p K^{pT}}{\sqrt{d_k}}\right) V^p \tag{2}$$

where $\sigma$ is active function, $P$ is the view type set, $p \in P$ is a view and $\Omega^p$ is a learnable view type weight matrix. $K^{pT}$ is the transpose matrix of view $p$ key matrix, and $d_k$ is the variance of $Q$ and $K$. In addition, our model assumes that a single $q_i$ and $k_i$ satisfy the mean of 0 and the variance of 1. Considering the more general case, $q_i$ and $k_i$ satisfy the mean value of 0 and the variance is $\sigma$, then $D(q_i k_i^T) = \sigma^4$. And $D(QK^T) = d_k \sigma^4$. In any case, divide by $\sqrt{d_k}$ to ensure that $D(QK^T) = D(q_i k_i^T)$. The reason to guarantee this is to make softmax not affected by the dimension of the vector. Furthermore, after adding multi-head attention structures, the embedding matrix can be formulated as:

$$\begin{cases} \text{head}_i = \text{Attention}(Q_i, K_i, V_i) \\ \text{Head}_i = \text{Concat}(\text{head}_1, \text{head}_2, \dots, \text{head}_n)W^o \end{cases} \tag{3}$$

where $W^o$ is the weight matrix of each head. Therefore, we can get the hidden nodes and edges features embedding matrix:

$$\begin{cases} H(X_V) = \text{Concat}(h_1(X_V), h_2(X_V), \dots, h_n(X_V))W_V^o \\ H(X_E) = \text{Concat}(h_1(X_E), h_2(X_E), \dots, h_n(X_E))W_E^o \end{cases} \tag{4}$$

**Aggregation nodes and edges information.** For each molecular graph view, we use graph transformer to obtain all nodes and edges features. All nodes' features is $X_{v_i}, \forall v_i \in V$, and the all edge nodes' features are $X_{e_{ij}}, e_{ij} \in E$. PharmHGT interactively operates on edge hidden states $H(X_{e_{ij}})$, node hidden state $H(X_{v_i})$, message $M_V(X_{v_i})$ and $M_E(X_{e_{ij}})$. To learn different knowledge from multi-view snapshots, we build a view attention message passing strategy that is based on multi-head attention structures, the node and edge feature are propagated at each iteration, $t$ denotes the current depth of the message passing, each step proceeds as follows:

$$M_V^1(X_{v_i}) = \sum_{\Theta_{\mathcal{N}(v_i)}} H(X_{\Theta_{\mathcal{N}(v_i)}}), t = 1 \tag{5}$$

$$M_E^1(X_{e_{ij}}) = H(X_{v_i}), t = 1 \tag{6}$$

$$M_V^t(X_{v_i}) = \sum_{\Theta_{\mathcal{N}(v_i)}} \text{Attention}\left(H^{t-1}(X_{v_i})W_{v_i}^Q, \right.$$
$$\left. M^{t-1}(X_{\Theta_{\mathcal{N}(v_i)}})W_{m_i}^K, H^{t-1}(X_{v_i})W_{v_i}^V\right), t > 1 \tag{7}$$

$$M_E^t(X_{e_{ij}}) = \text{Linear}\left(M_E^1(X_{e_{ij}}) + H^t(X_{v_i}) \right.$$
$$\left. - H^{t-1}(X_{e_{ji}})\right), t > 1 \tag{8}$$

where the $\Theta_{\mathcal{N}}(v_i)$ is the function to find edges directed to node $v_i$. Considering of the vanishing gradient issue, we set a simple residual block to make module training more stable during multi-views message passing:

$$\begin{cases} H^t(X_{v_i}) = H^{t-1}(X_{v_i}) + M_V^t(X_{v_i}) \\ H^t(X_{e_{ij}}) = H^{t-1}(X_{e_{ij}}) + M_E^t(X_{e_{ij}}) \end{cases} \tag{9}$$

**Fusion multi-views information.** For a given molecule, we obtain all types of representations of the three views of molecule atom-level, pharm-level, and junction-level by the above steps. Besides, the Gated Recurrent Unit is applied as a vision readout operator to get all three views feature vector $\{Z_\alpha, Z_\beta, Z_\gamma\}$ of the molecule, where $Z_\alpha$ is the vector of atom-level view, $Z_\beta$ is the vector of pharm-level view and $Z_\gamma$ is the vector of junction-level view.

Then, the acquired three views features are aggregated to the final features through the attention layer again, and the final representation vector of a molecule

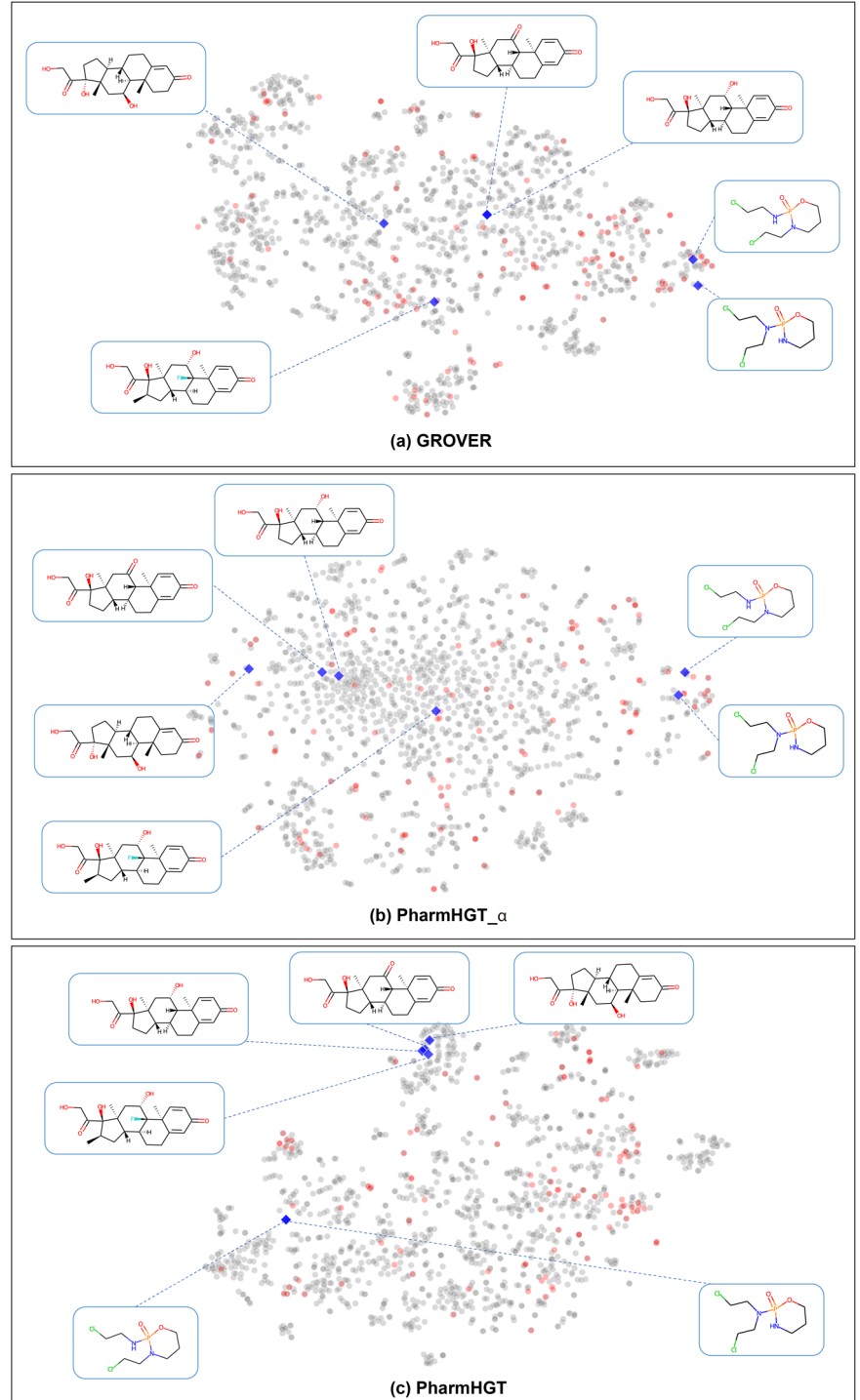

**Fig. 5 Case study.** Case study by t-SNE visualization of molecular features on ClinTox dataset from **a** GROVER, **b** PharmHGT_α, and **c** PharmHGT. Where molecular with toxicity are colored in gray, non-toxic molecules are in red and blue indicating six molecules are selected for the case study that showed toxicity in clinical trials and is still toxic after marketing.

is obtained. The readout attention function is:

$$\mathrm{ReadOutAttention}(X, Y) = \sigma\left(\frac{X \cdot Y^T}{\sqrt{d_k}}\right)X \qquad (10)$$

Specifically, the pharm-level-based contains the features of the reaction information, and we first aggregate it with the junction-level-based features to capture the associated information of pharmacophores and atoms and the reaction

information between pharmacophores. The formula is as follows:

$$Z_{\gamma\beta} = \mathrm{ReadOutAttention}(Z_\gamma, Z_\beta) \qquad (11)$$

Then we are aggregating this information with atom-level-based feature information to obtain the final molecular global feature representation (Fig. 2). The attention layer can distinguish the importance of features and adaptively assign more weight to more important features.

$$Z = \mathrm{ReadOutAttention}(Z_\alpha, Z_{\gamma\beta}) \qquad (12)$$

Finally, we perform downstream property predictions $\hat{y} = f(Z)$ where $f(\cdot)$ is a fully connected layer for classification or regression.

## Data availability

All related data in this paper are public. All downstream datasets can be downloaded from MoleculeNet.

## Code availability

The implementation of PharmHGT is publicly available at https://github.com/stardj/PharmHGT/.

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

## Acknowledgements

This study was supported in part by the ERC IMI (101005122), the H2020 (952172), the MRC (MC/PC/21013), the Royal Society (IEC/NSFC/211235), and the UKRI Future Leaders Fellowship (MR/V023799/1).

## Author contributions

Y.J., S.J., and Z.N. led the research. Y.J., S.J., X.J., and G.Y. contributed technical ideas. S.J., Y.J., and W.W. developed the proposed method. S.J., Y.J., Z.N., and X.L. performed analysis. X.J., X.L., X.Z., X.X., Q.Z., and Z.N. provided evaluation and suggestions. All authors contributed to the manuscript.

## Competing interests

Y.J., X.J., X.X., W.W., and Z.N. were employees at MindRank AI Ltd. The other authors have no conflicts of interest.
