## [Peer Review File · Communications Chemistry]

Reviewers' comments:

Reviewer #1 (Remarks to the Author):

This paper designs a pharmacophoric-constrained heterogeneous graph to exploit pharmacophore information including functional groups and chemical reactions for molecular property prediction. BRICS is used to split molecules into fragments that retain the reaction information. A transformer based model is developed to propagate message on the basis of three views of this heterogeneous graph. Experimental evaluation shows that the proposed method is effective in improving accuracy of drug property prediction.

Strengths:

- The heterogeneous graph proposed to connect atom nodes and pharmacophore nodes seems new and is interesting.
- The model performs better than some existing methods on 7 given datasets, especially on the regression tasks.

Weaknesses:

Major issues:

- About novelty: to the best of my knowledge, there are works that uses both atom-level, fragment-level and junction level embedding for molecule property prediction, only the method of combing these information is different. So the novelty of this paper is limited.
- About fusion multi-view information: First, they aggregate pharm-level-based features with junction-level-based features, then with atom-level-based features. Why? Can the authors explain their intuition or reason behind doing so. It would be better to try on other aggregation schemes with the three views, and check their effects on the model.
- In Table 2, they show the comparisons with some existing methods. Although PharmHGT performs best on the 7 datasets, the improvement is not significant.
- More datasets should be used for evaluation, such as bace and hiv. Some recently proposed models like Graphormer are not compared in this paper.

Minor issues:

- The description of the Heterogeneous Graph Transformer model in Fig.2 is confusing. It is not clear whether the following process is applied to the complete heterogeneous graph or to the three views separately.
- Reasons should be given for the use of the readout attention function in the part of Fusion Multi-views Information. More explanations about this module are preferred.

Reviewer #2 (Remarks to the Author):

Reviewer (JJT)

Basic reporting

1. Abstract:

The authors excessively described the research significance. However, they describe design PharmHGT, or the study significantly show the performance of the PharmHGT? or Testing the design? No results found on the molecular property prediction. Superior performance of the design or experimental results? No results are added or found in the Abstract.

2. Introduction

References 1-5, 6-7, 8-10 considered old. Authors has to refer the RECENT work on GNN and its

VARIANCE, before identify the GAP in the research using GNN parameters!
HGT itself a variance of GNN... Suggest the authors read more prediction works and demonstrate the significance of molecule prediction using GNN and its variance. for example,
Predicting Drug-Target Interactions With Multi-Information Fusion
DeepNC: a framework for drug-target interaction prediction with graph neural networks.
Deep Learning in Drug Target Interaction Prediction: Current and Future Perspectives
Drug-target interaction prediction via multi-channel graph neural networks

No Source Code is available. The Git Link is Empty "coming soon" message
Hard to evaluate the work and contribution.

There is NO significant difference with HGT of Claimed PharmHGT.... (Proposed work)

Figure 1 did not explain clearly Pharm-level view and its valences of the molecules and message passing.

3. Notation and Problem Definition

Definition 1, 2,3 represented the graph views. However, it is not clear why it use BRICS and message passing and attention mechanism/ reaction features are carried out between molecules (Mathematical equations are missing) Suggest the authors must understand the use of BRICS just to do the fragmentation. E.g. [Idea extraction from molecular graphs, the molecule fragmentation method that leverages a retrosynthesis-based algorithm BRICS]

4. Results

Expected to have discussion of experimental result rather dataset description in appendix.

Reference [13]

DGL-LifeSci is a GNN-based modeling on custom datasets for molecular
a) property prediction, b) reaction prediction, and c) molecule generation

Out of 7 benchmark datasets, use for two different task type. (Classification, regression)

Does, Fig2 downstream tasks predict C/R.

- not clear on this purpose of this.

- Fig 2 says mpp, (molecular property prediction)

Fig 2 has beautifully colorful. However, there is no significant indication written in the text and there is no support of GitHub Code given. Hard to evaluate this!!

If the authors propose only the framework in Figure 2, the article must justify from input and HGT execution or PharmHGT and the output with good discussion.

Table 2

What was the iterations?

Task compiled as in a Single Code or the datasets are compiled in different codes with different dataset?

It shows the best performance of PharmHGT,

Does PharmHGT has used all the 7 molecular property dataset and compared?

Do you have algorithm/pseudo code for PharmHGT?

check spelling Section 3.4

The article has to go through a proper proof reading, and order of sections.

Fig 4 is the result? it shows the visual output but no experimental evaluation.

Fig 5 PharmHGT alpha, no prior explanation of this..

Fig 5 visualize molecules in terms of classification? PharmHGT_alpha extracted Toxic21 dataset? how it is extracted and mapped?

Section 3 as RESULTS

Section 4 as related work

Section 5 as conclusion

Section 6 Methods

The above sequence is very confusing. Suggest Authors must mandatory follow the Springer nature template.

Section 3.7 starts with case study and how did you get the figure 5? Is there any connection to Table 2? There is no training and testing strategies of the curved plots? Suggest the authors must add these plots with all the 7 benchmark datasets learning and molecular prediction ROC

Reviewer #3 (Remarks to the Author):

The manuscript entitled "Pharmacophoric-constrained heterogeneous graph transformer" by Jiang and coauthors presents a design, called Pharmacophoric-constrained Heterogeneous Graph Transformer (PharmHGT), which uses multi-views molecular representation graphs based on graph neural networks (GNN). PharmHGT represents molecules and extracts chemical information including functional substructures for classification and regression purposes. The graph theoretical descriptions are well described, and the results show improvements over previous methods. However, the manuscript requires substantial refinements and additions, and the source code must be provided for the work to be properly reviewed. Please consider the following suggestions:

1. There is no code in the link provided as repository.
2. Abstract, missing "s" in features: can better capture the pharmacophoric structure and chemical information feature.
3. Introduction, add "information": maps the molecular information to a feature vector.
4. Introduction, add representative references for the conventional methods.
5. Add "work": Recently, Zhang et al. [7, 11] work has begun to focus.
6. Add a reference to SMILES.
7. Use labels for panels in Figure 1. Please improve all the legends for figures and tables, those should include sufficient information for each item to be understood by itself.
8. Table 1. Add a legend to better describe the table and clarify the column names: add the symbol "#" for Molecules and specify the meaning of Task.
9. Table 2. Specify what the performance metric is.
10. Add references for the datasets used.
11. Please explain how the model is able to perform both classification and regression. Is it the same model or 2 separate models with the same architecture but trained independently?
12. More details regarding implementation should be provided in the Methods. For example, what is the format of the input? SMILES?
13. Add a reference and a short description about the meaning and objective of the ablation study to clarify that it refers to the neural network behavior (not a procedure with a biological sample).
14. Figure 3 needs a legend, include what the compared models mean.
15. Could you add a brief comparison of the compared methods in terms of computing resources

(training time, memory, etc.)? How much less is the computational cost of PharmHGT?
16. Consider placing the Related work section after the Introduction.

1. Reviewer #1

RC: *This paper designs a pharmacophoric-constrained heterogeneous graph to exploit pharmacophore information including functional groups and chemical reactions for molecular property prediction. BRICS is used to split molecules into fragments that retain the reaction information. A transformer based model is developed to propagate message on the basis of three views of this heterogeneous graph. Experimental evaluation shows that the proposed method is effective in improving the accuracy of drug property prediction.*

1.1. Strengths

RC: *- The heterogeneous graph proposed to connect atom nodes and pharmacophore nodes seems new and is interesting.
- The model performs better than some existing methods on 7 given datasets, especially on the regression tasks.*

1.2. Weaknesses

1.2.1 Major issues

RC: *About novelty: to the best of my knowledge, there are works that use both atom-level, fragment-level, and junction-level embedding for molecule property prediction, only the method of combing this information is different. So the novelty of this paper is limited.*

AR: Thanks for this comment, and during the study design stage, we conducted a comprehensive literature review. Currently, there were some studies using the fragment of molecules and adding such information to the model [1,2]. The novelty of our study is shown in the following points: 1) We further consider the information of chemical reaction and pharmacophoric information; 2) Our model learns the representations not only from the single view, we build the multi-views heterogeneous graph for each molecule and we conduct the message passing in inter-views and message attention aggregation intra-view, respectively. This process enables our model capable to learn the information from the interaction among atoms, fragments, and molecules.

[1] Zhang Z, Guan J, Zhou S. FraGAT: a fragment-oriented multi-scale graph attention model for molecular property prediction[J]. Bioinformatics, 2021, 37(18): 2981-2987.

[2] Zhang Z, Liu Q, Wang H, et al. Motif-based graph self-supervised learning for molecular property prediction[J]. Advances in Neural Information Processing Systems, 2021, 34: 15870-15882.

Fig.3 Ablation results on BBBP and ESOL datasets. The "X" represent the PharmHGT, the "X_" represent different PharmHGT variants of aggregating atom-level, junction-level, and pharm-level features.

RC: *About fusion multi-view information: First, they aggregate pharm-level-based features with junction-level-based features, then with atom-level-based features. Why? Can the authors explain their intuition or reason behind doing so. It would be better to try on other aggregation schemes with the three views, and check their effects on the model.*

AR: Common molecular property predictions models are based on atomic-level features (α), so intuitively consider the atomic-level features as the basis, and aggregate the features of the other two perspectives on top of it, and the junction-level (γ) is the connection relationship between the pharm-level (β) and the atom-level. So first aggregate pharm-level-based features with junction-level-based features ($\gamma\beta$), then with atom-level-based features ($\gamma\beta\alpha$). There are 6 random aggregation schemes for these three views, namely: $\gamma\beta\alpha$, $\beta\gamma\alpha$, $\alpha\gamma\beta$, $\gamma\alpha\beta$, $\beta\alpha\gamma$, $\alpha\beta\gamma$. Because junction-level (γ) is the connection relationship, it cannot be used as the final basic aggregation feature, so $\beta\alpha\gamma$, $\alpha\beta\gamma$ is not feasible. It can be seen from Fig.2 that the feature matrix dimensions of $\beta\gamma\alpha$ and $\gamma\alpha\beta$ do not match, and $\gamma\beta\alpha$ is the aggregation scheme of the PharmHGT model. So we have added PharmHGT model variants (PharmHGT_ $\gamma\alpha\beta$: by aggregating features of the junction-level graph with the atom-level graph, then to the pharm-level graph.) in the ablation experiment part. The result is shown in the figure below. Experimental results show that the PharmHGT performs best with existing information aggregation schemes when three views work together.

RC: *In Table 2, they show the comparisons with some existing methods. Although PharmHGT performs best on the 7 datasets, the improvement is not significant.*

AR: Thanks for the comment. Our model provides a new perspective on molecular property prediction. Because the models with better performance in the baseline model are basically pre-trained models, such models need to consume a lot of computing resources and take more time during the training process. Our model outperforms these models without pre-training, demonstrating the potential of our model. In future work, we will try pre-training schemes to further improve the accuracy of predictions more significantly.

RC: *More datasets should be used for evaluation, such as bace and hiv. Some recently proposed models like Graphormer are not compared in this paper.*

AR: Thanks you for this kind comment. We have added more experiments with the mentioned two datasets in our works. The Graphormer model is a 3D model which requires the 3D conformation information of small molecules. However, the data set for the molecular property prediction task does not provide 3D conformation. In order for the model to perform normal training and prediction, we use rdkit to get 3D conformation of each data set for training and testing with a random force field, and the final result is illustrated in Table R1 and Table R2. Accordingly, there are some inconsistencies between our model and Graphormer in their target tasks and the inputs. For better comparability, we did not put the Graphormer into the benchmark and we select other eight benchmarks for performance comparison.

Table R1: Overall Performance comparison to the state-of-the-art methods on molecular property prediction classification tasks. The results of baselines are obtained by us using a 5-fold cross validation with scaffold split and doing experiment on each task for one time. The values in this table are the Mean and standard deviation of ROC-AUC values.

Classification(ROC-AUC%, higher is better↑)						
Dataset	BBBP	BACE	ClinTox	Tox21	SIDER	HIV
Molecules	2039	1513	1478	7831	1427	41127
Task	1	1	2	12	27	1
Splitting strategy	Scaffold	Scaffold	Scaffold	Scaffold	Scaffold	Scaffold
Graphormer	93.6(2.41)	83.3(1.16)	88.1(3.80)	80.8(2.00)	62.0(1.20)	78.9(0.91)
PharmHGT	95.4(1.15)	86.5(2.21)	94.5(0.42)	83.9(0.56)	66.9(1.63)	80.6(0.21)

Table R2: Overall Performance comparison to the state-of-the-art methods on molecular property prediction regression tasks. The results of baselines are obtained by us using a 5-fold cross validation with scaffold split or Random split and doing experiment on each task for one time. The values in this table are the Mean and standard deviation of RMSE values.

Regression(RMSE, lower is better↓)						
Dataset	ESOL	FreeSolv	Lipophilicity	ESOL	FreeSolv	Lipophilicity
Molecules	1128	642	4200	1128	642	4200
Tasks	1	1	1	1	1	1
Splitting strategy	Random	Random	Random	Scaffold	Scaffold	Scaffold
Graphormer	0.858(0.032)	1.652(0.210)	0.839(0.020)	0.931(0.042)	2.019(0.752)	1.097(0.389)
PharmHGT	0.680(0.137)	1.266(0.239)	0.583(0.026)	0.839(0.049)	1.689(0.516)	0.638(0.040)

1.2.2 Minor issues

RC: *The description of the Heterogeneous Graph Transformer model in Fig.2 is confusing. It is not clear whether the following process is applied to the complete heterogeneous graph or to the three views*

Algorithm 1 algorithm caption

Input:

Graph $G = \{V, E\}$;
view types $t \in \mathcal{T}$;
Node features X_{v^t} , where $v^t \in V^t$;
Edge features $X_{e_{v,u}^t}$, where $e_{v,u}^t \in E^t$.

Ensure:

Neighbors find function $\Theta_{\mathcal{N}}$.
Readout attention function $\Theta_{\mathcal{R}}$.
Index inverse function $\Theta_{\mathcal{I}}$.

```
1: for  $t$  in  $\mathcal{T}$  do  
2:    $\mathcal{H}_0^t(E^t) \leftarrow X_{E^t}, \mathcal{H}_0^t(V^t) \leftarrow X_{V^t}$   
3:   for  $k = 1$   $K$  do  
4:      $\mathcal{M}_k^t(V^t) \leftarrow \mathcal{H}_{k-1}^t(E_{\Theta_{\mathcal{N}}^t(V^t)}^t)$   
5:      $\mathcal{H}_k^t(V^t) \leftarrow \text{Linear}(\text{Cat}(\mathcal{H}_{k-1}^t(V^t), \mathcal{M}_k^t(V^t)))$   
6:      $\mathcal{M}_k^t(E^t) \leftarrow \mathcal{H}_k^t(V^t) - \mathcal{H}_{k-1}^t(E_{\Theta_{\mathcal{I}}^t(V^t)}^t)$   
7:      $\mathcal{H}_k^t(E^t) \leftarrow \sigma(\mathcal{H}_0^t(E^t) + \mathcal{M}_k^t(E^t))$   
8:   end for  
9: end for  
10:  $\mathcal{H}(V) \leftarrow \text{Linear}(\text{Cat}(\Theta_{\mathcal{R}}(\mathcal{H}_K^T(V^T)), \Theta_{\mathcal{R}}(\mathcal{M}_K^T(V^T))))$   
11: return  $\mathcal{Z} \leftarrow \text{Readout}(\mathcal{H}(V))$ 
```

separately.

AR: We apologize for the unclear description in Fig.2. Our model applied to the complete heterogeneous graph, and we have revised Fig.2 to clearly illustrate the input of our model, and add the figure legend.

RC: *Reasons should be given for the use of the readout attention function in the part of Fusion Multi-views Information. More explanations about this module are preferred.*

AR: In each step, each view will first conduct the message passing independently, obtain the graph feature matrix of each view and input it to the readout module. In this module, the junction_view feature matrix will first do attention aggregation with the pharm_view feature matrix to obtain the aggregation feature matrix, then that matrix will do attention aggregation with the atom_view feature matrix, and finally obtain the features of each node, and input those nodes as a sequence into the GRU to get the representation vector of the entire small molecule. And we have added an iteration pseudo-code as an iteration detailed description in Appendix. It is as algorithm 1:

2. Reviewer #2

2.1. Abstract

RC: *The authors excessively described the research significance. However, they describe design PharmHGT, or the study significantly show the performance of the PharmHGT? or Testing the design? No results found on the molecular property prediction. Superior performance of the design or experimental results? No results are added or found in the Abstract.*

AR: Thanks for pointing this out. We have revised the abstract according to the comments suggest as: "Extensive downstream experiments prove that PharmHGT achieves remarkably superior performance over the state-of-the-art models (the performance of our model are up to 1.55% in ROC-AUC and 0.272 in RMSE higher than the best baseline model) on molecular properties prediction.", "Further visualization studies also indicated a better representation capacity achieved by our model".

2.2. Introduction

RC: *References 1-5, 6-7, 8-10 considered old. Authors has to refer the RECENT work on GNN and its VARIANCE, before identify the GAP in the research using GNN parameters! HGT itself a variance of GNN... Suggest the authors read more prediction works and demonstrate the significance of molecule prediction using GNN and its variance. for example, Predicting Drug-Target Interactions With Multi-Information Fusion DeepNC: a framework for drug-target interaction prediction with graph neural networks. Deep Learning in Drug Target Interaction Prediction: Current and Future Perspectives Drug-target interaction prediction via multi-channel graph neural networks*

AR: Thanks for pointing this out. We have readed more prediction works and added the description about the significance of molecule prediction using GNN and its variants in our work. We have added more description in the introduction section as "The Graph Neural Networks (GNNs) have gained increasingly more popularity due to its capability of modeling graph structured data. For the association prediction task of biological network data, the heterogeneous graph neural network algorithms [1-3] have achieved remarkable results. Molecules can be naturally expressed as a graph structure, so the GNNs method can effectively capture molecular structure information, including nodes (atoms) and edges (bonds) [4]." In addition, we also have added new references of molecular property prediction works based on GCN and GAT in the manuscript.

[1] Li Y, Qiao G, Wang K, et al. Drug-target interaction prediction via multi-channel graph neural networks[J]. Briefings in Bioinformatics, 2022, 23(1).

[2] Abbasi, K., Razzaghi, P., Poso, A., Ghanbari-Ara, S., Masoudi-Nejad, A.: Deep learning in drug target interaction prediction: current and future perspectives. Current Medicinal Chemistry 28(11), 2100-2113 (2021)

[3] Tran, H.N.T., Thomas, J.J., Malim, N.H.A.H.: Deepnc: a framework for drug-target interaction prediction with graph neural networks. PeerJ 10, 13163 (2022)

[4] Rong Y, Bian Y, Xu T, et al. Self-supervised graph transformer on large-scale molecular data[J]. Advances in Neural Information Processing Systems, 2020, 33: 12559-12571.

RC: *No Source Code is available. The Git Link is Empty "coming soon" message. Hard to evaluate the work and contribution.*

AR: Thanks for pointing out this, we have uploaded the usable code and data at <https://github.com/stardj/PharmHGT/>

RC: *There is NO significant difference with HGT of Claimed PharmHGT... (Proposed work)*

AR: Thanks for this comments. Compare with HGT, our model is differentiating in message aggregation and the readout process. The Heterogeneous Graph Transformer (HGT)[5] architecture is proposed for modeling Web-scale heterogeneous graph. The model use node- and edge-type dependent parameters to characterize the heterogeneous attention over each edge, empowering HGT to maintain dedicated representations for different types of nodes and edges. In general this model is to aggregate node information in a large network to obtain nodes representation. While in our model, we aggregate nodes information in multiple views of the molecule

in order to obtain a feature representation of the entire molecular graph.

[5] Hu Z, Dong Y, Wang K, et al. Heterogeneous graph transformer[C]//Proceedings of The Web Conference 2020. 2020: 2704-2710.

RC: *Figure 1 did not explain clearly Pharm-level view and its valences of the molecules and message passing.*

AR: The Fig.1 is an example to overview of the molecular segmentation process and the construction of heterogeneous molecular graph. We have revised the Fig.2 to help reader understand the architecture of our heterogeneous graph and have added the legend. “Firstly, the heterogeneous molecular graph is formalized as the feature matrix. Then, the feature matrix of each view will first do message passing independently to obtain the graph feature matrix of each view. Finally, the junction_view feature matrix will first do attention aggregation with the pharm_view feature matrix to obtain the an aggregation feature matrix, then that matrix will do attention aggregation with the atom_view feature matrix, and finally obtain the features of each node, and input those node as a sequence into the GRU to get the representation vector of the entire small molecule.”

2.3. Notation and Problem Definition

RC: *Definition 1, 2,3 represented the graph views. However, it is not clear why it use BRICS and message passing and attention mechanism/ reaction features are carried out between molecules (Mathematical equations are missing) Suggest the authors must understand the use of BRICS just to do the fragmentation. E.g. [Idea extraction from molecular graphs, the molecule fragmentation method that leverages a retrosynthesis-based algorithm BRICS]*

AR: There are three wide-used compound segmentation methods: random segmentation, recap segmentation and BRICS segmentation. Random segmentation does not necessarily conform to chemical rules. Recap segmentation in rdkit has no information about chemical bond breaking, while BRICS segmentation is performed through 46 defined segmentation methods [1]. This segmentation method is in line with the chemical reaction as the basis of breaking the bond and the segmentations by BRICS can provide more information for us to build a pharm_level graph. Accordingly, we adopt this method to perform the segmentation.

[1] Degen J, Wegscheid-Gerlach C, Zaliani A, et al. On the Art of Compiling and Using ‘Drug-Like’ Chemical Fragment Spaces[J]. ChemMedChem: Chemistry Enabling Drug Discovery, 2008, 3(10): 1503-1507

2.4. Results

RC: *Expected to have discussion of experimental result rather dataset description in appendix. Reference [13] DGL-LifeSci is a GNN-based modeling on custom datasets for molecular a) property prediction, b) reaction prediction, and c) molecule generation*

AR: Thanks for pointing this out. We have add the analysis and discussion of experiments in the Appendix, including the experimental setting, computational cost of PharmHGT, etc (see Appendix). The Reference [13] was misplaced, we apologize for this inconvenience and we have revised the reference of the datasets source.

RC: *Out of 7 benchmark datasets, use for two different task type. (Classification, regression) Does, Fig2 downstream tasks predict C/R. - not clear on this purpose of this. - Fig 2 says mpp, (molecular property prediction)*

AR: The task of molecular property prediction includes classification and regression tasks. For datasets of different tasks, we have obtained the feature of molecules to predict the properties of molecules, and predicted the specific category(classification) or an attribute value (regression). we have revised the Fig.2 to help reader

understand our model.

RC: *Fig 2 has beautifully colorful. However, there is no significant indication written in the text and there is no support of GitHub Code given. Hard to evaluate this!! If the authors propose only the framework in Figure 2, the article must justify from input and HGT execution or PharmHGT and the output with good discussion.*

AR: Thanks for your comments. We have added the figure legends and published all the code on github: github.com/stardj/PharmHGT, and we revised the Fig.2 to describe our model more clear.

2.5. Table 2

RC: *What was the iterations?*

AR: In each step, each view will first do message passing independently, obtain the graph feature matrix of each view and input it to the readout module. In this module, the junction_view feature matrix will first do attention aggregation with the pharm_view feature matrix to obtain the aggregation feature matrix, then that matrix will do attention aggregation with the atom_view feature matrix, and finally obtain the features of each node, and input those nodes as a sequence into the GRU to get the representation vector of the entire small molecule. After obtaining the representation vector, the classification and regression tasks can be performed through the parallel neural network layers. And we will add an iteration pseudo code as iteration detailed description in Appendix.

RC: *Task compiled as in a Single Code or the datasets are compiled in different codes with different dataset?*

AR: For all datasets, models are compiled as in a single code with different input datasets. After obtaining the feature vector of molecules, two parallel neural network layers are used to predict the classification and regression tasks.

RC: *It shows the best performance of PharmHGT, Does PharmHGT has used all the 7 molecular property dataset and compared?*

AR: Yes we did, and all the code have been published on github:<https://github.com/stardj/PharmHGT>, the datasets, training stratege and evaluate method are consistent with the benchmark in a fair comparison.

RC: *Do you have algorithm/pseudo code for PharmHGT?*

AR: Thank you for your comments, We add an iteration pseudo-code for our model in Appendix.

2.6. check spelling Section 3.4

RC: *The article has to go through a proper proof reading, and order of sections.*

AR: Thank you for your kind reminder, we will go through a proper proof reading and fix all the format mistake.

RC: *Fig 4 is the result? it shows the visual output but no experimental evaluation.*

AR: Thanks for your comments. For this result, we define all molecules with a label of 0 as nontoxic compounds, and any molecule with a label of 1 as a toxic compound, and molecules with similar toxicity tend to have more similar feature spaces. Therefore, we visualize their embeddings by t-SNE, and evaluate whether the model can learn effective molecular representations by whether the toxic and non-toxic molecules have a clear boundary. We have added the description in section 4.6.

RC: *Fig 5 PharmHGT alpha, no prior explanation of this.*

AR: We describe the PharmHGT alpha in 3.5 part, and conducted ablation studies with it.

RC: *Fig 5 visualize molecules in terms of classification? PharmHGT_alpha extracted Toxic21 dataset? how it is extracted and mapped?*

AR: Fig.5 visualized molecular features from ClinTox dataset, we think it would be more valuable if our model could perform better on clinical toxicity. We choose 6 specific toxicity compounds for a case study, those compounds come from two different types, one is natural products and the other is artificially designed products. Although the molecules of each type is different, their toxic pharmacophores are highly consistent. We do this case study to show that our model has better-learned pharmacophore information than other benchmark models.

RC: *Section 3 as RESULTS Section 4 as related work Section 5 as conclusion Section 6 Methods The above sequence is very confusing. Suggest Authors must mandatory follow the Springer nature template.*

AR: Thanks for pointing this out. We have changed the section sequence follow the Springer nature template.

RC: *Section 3.7 starts with case study and how did you get the figure 5? Is there any connection to Table 2? There is no training and testing strategies of the curved plots. Suggest the authors must add these plots with all the 7 benchmark datasets learning and molecular prediction ROC*

AR: Fig.5 visualized compounds from the ClinTox dataset with their trained embeddings by t-SNE, and the result of our model is illustrated on Table2. We have described our training and evaluation strategies on 3.2.2, and we have added prediction ROC curved plots in Appendix. The result shown as Fig.A1 and Fig.A2:

Fig.A1 Overall ROC curves on molecular property prediction classification tasks.

Fig.A2 ROC curves on each molecular property prediction classification tasks.

3. Reviewer #3

RC: *The manuscript entitled "Pharmacophoric-constrained heterogeneous graph transformer" by Jiang and coauthors presents a design, called Pharmacophoric-constrained Heterogeneous Graph Transformer (PharmHGT), which uses multi-views molecular representation graphs based on graph neural networks (GNN). PharmHGT represents molecules and extracts chemical information including functional sub-structures for classification and regression purposes. The graph theoretical descriptions are well described, and the results show improvements over previous methods. However, the manuscript requires substantial refinements and additions, and the source code must be provided for the work to be properly reviewed. Please consider the following suggestions:*

AR: We thank the Reviewer for the great summary and the constructive feedback on this study. We have revised the manuscript according to those comments.

3.1. Minor Points

RC: *1. There is no code in the link provided as repository.*

AR: Thanks for pointing this out. We have open-sourced our model, the code is available: <https://github.com/stardj/PharmHGT>

RC: *2. Abstract, missing "s" in features: can better capture the pharmacophoric structure and chemical information feature.*

AR: Thanks for pointing this out. We have revised the manuscript according to the comments suggest.

RC: *3. Introduction, add "information": maps the molecular information to a feature vector.*

AR: Thanks for pointing this out. We have revised the manuscript according to the comments suggest.

RC: 4. Introduction, add representative references for the conventional methods.

AR: Thanks for pointing this out. We have added representative references for the conventional methods.

RC: 5. Add “work”: Recently, Zhang et al. [7, 11] work has begun to focus.

AR: Thanks for pointing this out. We have revised the manuscript according to the comments suggest.

RC: 6. Add a reference to SMILES.

AR: Thanks for pointing this out. We have added the references.

RC: 7. Use labels for panels in Figure 1. Please improve all the legends for figures and tables, those should include sufficient information for each item to be understood by itself.

AR: Thanks for this suggestion. We have revised these legends for better understanding by readers.

RC: 8. Table 1. Add a legend to better describe the table and clarify the column names: add the symbol “#” for Molecules and specify the meaning of Task.

AR: Thanks for this suggestion. We have add the symbol “#” for Molecules and specify the meaning of Task.

RC: 9. Table 2. Specify what the performance metric is.

AR: For the regression tasks, root mean squared error (RMSE) is used as the metric. RMSE is a typical indicator of a regression model, which is used to indicate how much error the model will make in the prediction, and for larger errors, the weight is higher. So RMSE is the smaller the better. And for the classification, area under the receiver operating characteristic curve (ROC-AUC) is used. ROC-AUC is a typical metric for classification models and is used to measure the probability that the model classifies the molecular class correctly. ROC-AUC So is the larger the better.

RC: 10. Add references for the datasets used.

AR: We add the references for the datasets, "In this section, we present the proposed PharmHGT model and its results for molecular property prediction on seven datasets, these experiments datasets are from Wu et al. [26], including four classification and three regression tasks."

RC: 11. Please explain how the model is able to perform both classification and regression. Is it the same model or 2 separate models with the same architecture but trained independently?

AR: The model is mainly used to learn the representation of molecules. After obtaining the representation vector, the classification and regression tasks can be performed through the parallel neural network layers.

RC: 12. More details regarding implementation should be provided in the Methods. For example, what is the format of the input? SMILES?

AR: Yes, the input data is in SMILES format, but for our model, we preprocess each smile to the graph as input. "The inputs of PharmHGT is the feature matrix of the node X_V and the feature matrix of edge X_E , the features of all nodes can be obtained according to the intensity of the attention between the node and the related edge." Details of the construction of the feature matrix are given in the Appendix.

RC: 13. Add a reference and a short description of the meaning and objective of the ablation study to clarify that it refers to the neural network behavior (not a procedure with a biological sample).

Table S6: Overall Computing resources performance comparison to the state-of-the-art methods on ESOL datasets. The results of baselines are obtained by us using a 5-fold cross validation with scaffold split and doing experiment on each task for one time.

Models	Params	Pretrain	Trainng time(s/epoch)	Training cost(s/fold)
AttentiveFP	0.65M	No	0.41	20.5
FragGAT	1.75M	No	0.72	36.0
MGSSL	2.72M	Yes	1.25	62.5
MPNN	1.06M	No	0.60	30.0
DMPNN	1.62M	No	0.80	40.0
CMPNN	2.00M	No	0.83	25.0
CoMPT	2.60M	No	1.80	72.0
GROVER _{base}	48.0M	Yes	4.79	115
GROVER _{large}	107M	Yes	5.83	140
PharmHGT	2.50M	No	1.20	24.0

AR: Our ablation experiments are mainly to explore the effect of using different view features. When using a single-view feature, the network architecture of the model is the same. This result shows that when using a single-view feature, only the atom-level view is the best. We also explore the performance of combination schemes using different two-view features when using two-view features. However, the results of using a single view feature or two view features are lower than those of the scheme that comprehensively considers three view features. This also shows that the multi-view representation method we proposed can learn the characteristic information of molecules more effectively. In the section "Case Study", we also proved that adding pharm-level and junction-level can better learn pharmacophore information than other benchmark models.

RC: *14. Figure 3 needs a legend, include what the compared models mean.*

AR: Thanks for pointing this out. We have added the legend of Figure 3. The "X" represent the result of PharmHGT, the "X_" represent different PharmHGT variants of aggregating atom-level view, junction-level view, and pharm-level view features.

RC: *15. Could you add a brief comparison of the compared methods in terms of computing resources (training time, memory, etc.)? How much less is the computational cost of PharmHGT?*

AR: Thank you for your kind reminder, we have added those in Appendix. The result shown as Table S6:

RC: *16. Consider placing the Related work section after the Introduction.*

AR: Thanks for this suggestion. We have placed the Related work section after the Introduction.

REVIEWERS' COMMENTS:

Reviewer #2 (Remarks to the Author):

The authors had extensively revised the article.

I am happy for the major revision for the manuscript with all the sixteen comments were addressed and amended.

Reviewer #3 (Remarks to the Author):

The authors have addressed the reviewers' concerns and suggestions.